# Advances in the Prenatal Management of Fetal Cardiac Disease

**DOI:** 10.3390/children9060812

**Published:** 2022-05-31

**Authors:** Chetan Sharma, Joseph Burns, Krittika Joshi, Monesha Gupta, Harinder Singh, Arpit Agarwal

**Affiliations:** 1Division of Pediatric Cardiology, Children’s Hospital of San Antonio, Baylor College of Medicine, San Antonio, TX 78207, USA; krittika.joshi@bcm.edu (K.J.); monesha.gupta@bcm.edu (M.G.); harinder.singh@bcm.edu (H.S.); arpit.agarwal@bcm.edu (A.A.); 2Division of Pediatrics, Cohen Children’s Medical Center, New York, NY 11004, USA; jburns9@northwell.edu

**Keywords:** prenatal intervention, congenital heart diseases

## Abstract

Advances in the field have improved the prenatal management of cardiovascular diseases over the past few decades; however, there remains considerable challenges in the approach towards patient selection as well as the applicability of available therapies. This review aims to discuss the current knowledge, outcomes and challenges for prenatal intervention for congenital heart disease.

## 1. Introduction

Among various congenital malformations diagnosed in utero, heart defects are one of the most common anomalies detected on prenatal ultrasound [1]. Fetal cardiac screening between 18–22 weeks of gestation is the standard of care, specifically in high risk pregnancy [2]; however, first trimester imaging is being offered by some centers. Over the last few decades, our ability to detect various congenital heart defects has improved tremendously, secondary to improvement in ultrasound machines, curvilinear probes, three-dimensional and 4D ultrasound etcetera. These advances in fetal imaging, better understanding of the medical pharmacokinetics and the impact of medications on fetal growth, as well as advances in prenatal transcatheter intervention, have allowed us to take steps towards performing various non-invasive and invasive fetal cardiac interventions.

Although advances in fetal imaging have contributed greatly to our ability to intervene prenatally, prenatal approaches to treat cardiac disease date back to 1975, when Eibschitz et al. reported the use of propranolol to treat fetal ventricular tachycardia [3]. In 1991, Maxwell et al. reported the first two cases of balloon valvuloplasty of the aortic valve to treat fetuses with critical aortic stenosis [4]. The role of fetal cardiac therapies vary from the non-invasive medical management of arrhythmias, to performing palliative procedures to allow fetuses to be delivered closer to term and/or improve survival probability in the fetuses who are at risk for fetal/neonatal demise.

Table 1 and Table 2 gives an overview of various fetal cardiac conditions for which fetal cardiac intervention as well as pharmacotherapy is commonly utilized.

## 2. Diagnosis and Management of Fetal Arrhythmia

Fetal arrhythmia can be broadly classified in two broad categories, namely, fetal bradycardia or fetal tachyarrhythmias. Using M-mode echocardiography and spectral Doppler, it is possible to evaluate sequential motion of atria and ventricle and thus evaluate the mechanism of arrhythmia.

### 2.1. Fetal Bradycardia

Fetal bradycardia can be classified as sinus bradycardia, bradycardia in the setting of long QT Syndrome and bradycardia in the setting of atrioventricular dissociation. The management of fetal bradycardia depends on the cause of bradycardia as well as the ventricular rate and presence or absence of hydropic changes. Fetal sinus bradycardia is defined as the fetal ventricular rate of 110 beats per minute or less in the setting of 1:1 atrioventricular association [5].

Bradyarrhythmias can occur from atrioventricular block from congenital abnormality of the atrioventricular node with congenital heart disease such as congenitally corrected transposition of the great arteries, heterotaxy syndrome or from channelopathies, including long QT mutations [6]. Acquired fetal bradyarrhythmias can occur from maternal antibodies to Ro (SS-A) and La (SS-B) [7]. Fetal bradycardia can be observed with blocked atrial premature complexes as well. In the setting of persistent fetal bradycardia, the potential consequences include fetal hydrops or fetal cardiomyopathy.

In the current era, various transplacental treatment options for immune-mediated atrioventricular block include the administration of fluorinated steroids such as dexamethasone, hydroxychloroquine and/or sympathomimetic agents [8,9,10].

More recently, there has been a spate of literature reporting the use of immunomodulatory therapies to prevent either the development or progression of the atrioventricular block to the complete atrioventricular block. Jaeggi et al. evaluated role of dexamethasone and beta-stimulation in 37 fetuses (25.6 +/− 5.2 gestational weeks) with complete heart blocks. In their study, 21 fetuses treated with dexamethasone were to found to have an improved one-year survival of 90%, compared with 46% without glucocorticoid therapy (P 0.02) [9]. In another study, Costedoat-Chalumeau et al. reported adverse events associated with use of dexamethasone. This study evaluated 13 pregnancies in 7 mothers with a history of prior babies with congenital heart blocks associated with maternal anti-SSA/Ro antibodies. In their experience, six pregnancies in which women were treated with dexamethasone (4–5 mg/day) had adverse outcomes, including spontaneous abortion (N:2), stillbirths (N:2) and two live births with intrauterine growth restriction and mild adrenal insufficiency [11].

A prior multicenter open-label study by Friedman et al. evaluated the role of IVIG in preventing the progression to second- or third-degree heart block with a small sample size, and demonstrated no benefit of IVIG to prevent the recurrence of heart block or reduce maternal antibody titers [12].

Carpenter et al. presented their experience in 1986 with fetal ventricular pacing for hydrops secondary to the complete atrioventricular block [13]. There are various ongoing animal studies evaluating the use of percutaneously implantable fetal pacemaker with mixed results [14].

### 2.2. Fetal Tachyarrhythmias

Fetal tachycardia was first described by Hayman et al. in 1930 [15]. Fetal tachycardia is defined as a fetal heart rate greater than 180–200 beats per minute [16]. Fetal tachyarrhythmias include supraventricular tachycardia (SVT) and atrial flutter, and ventricular tachyarrhythmia. The treatment for tachyarrhythmias primarily depends on the underlying mechanism for tachyarrhythmias. Various treatment options include transplacental therapy with antiarrhythmic such as digoxin, sotalol, amiodarone and flecainide [17].

In cases of severe hydrops, transplacental therapy is known to be less effective. There are case reports on the use of intraperitoneal and intra-amniotic amiodarone injections in the successful treatment of fetal atrial flutter [18].

## 3. Pharmacological Approach to Treatment of Intracardiac Rhabdomyomas

With the advances in fetal echocardiographic techniques, prenatal detection of cardiac tumors has enabled us to use transplacental pharmacotherapy to treat cardiac rhabdomyoma (Figure 1). Recent reports have highlighted the transplacental use of the mechanistic target of rapamycin (m-TOR) inhibitors such as sirolimus in the treatment of fetal rhabdomyomas causing outflow obstruction [19,20]. The results reported appear to be promising; however, there are no larger scale trial data to recommend routine use at this time.

## 4. Maternal Hyperoxia Therapy

Maternal hyperoxia therapy represents another frontier in fetal cardiac therapy. The role of maternal hyperoxygenation in fetuses with Hypoplastic Left Heart Syndrome (HLHS) has been matter of ongoing investigation. Various authors have proposed the role of maternal hyperoxygenation in increasing the fetal pO2 leading to an increase in pulmonary venous blood flow [21]. Increased pulmonary flow can in turn cause increased flow through fetal mitral and aortic valve and thus can have a potential role in “recruiting” the hypoplastic left ventricle [22].

Apart from potentially therapeutic effects, maternal hyperoxygenation can aid in the identification of fetuses with a lower degree of prenatal pulmonary vasoreactivity, thus identifying those with significant pulmonary vascular disease to predict a need for immediate postnatal intervention [23,24].

In a murine model, hyperoxia therapy demonstrated the reduction of excess trabeculation and muscular ventricular septal defects and improved the organization of coronary vasculature in genetically modified mice [25]. A small study of 24 mothers with fetuses known to have CHD demonstrated a variable response to hyperoxygenation on cerebral tissue oxygenation [26]. The authors admit that the small sample size limits the conclusions gleaned from the study, but transplacental oxygen delivery may represent a promising intervention for fetuses with complex CHD [26]. You et al. demonstrated improvement in cerebral blood oxygenation, particularly in fetuses with a single ventricle physiology in a small sample of 30 fetuses [27]. In a study including 43 fetuses, differential response based on the cardiac lesion, with an improved middle cerebral artery pulsatility index in response to maternal hyperoxygenation in patients with d-TGA, but not to a significant extent in cases with left- or right-sided obstructive lesions, was demonstrated [28].

As our understanding regarding maternal hyperoxia therapy is still evolving, its role and treatment protocols are variable. A systematic review of the current literature by Co-Vu et al. found that most protocols used 8 L with 100% oxygen by a non-rebreather at a mean gestational age of 33.4 weeks [29]. The duration of therapy ranges from 10 min in a single session to 3–4 h per day [30].

## 5. Percutaneous Fetal Cardiac Interventions

Use of percutaneous transcatheter fetal intervention is used in a few congenital heart diseases. In 1991, Maxwell et al. reported the first two cases of balloon valvuloplasty of the aortic valve to treat fetuses with critical aortic stenosis [4]. The specific examples include hypoplastic left heart syndrome (HLHS) patients with restrictive septum, HLHS patients with severe aortic valve stenosis, pulmonary atresia with intact ventricular septum, critical pulmonary stenosis and fetal hydrops with pericardial effusion [31].

As our learning curve as well as expertise increases, the outcomes of fetal intervention have demonstrated varying levels of success [32]. Larger registries have offered insight into the success of fetal intervention. The International Fetal Cardiac Intervention Registry, which included multicenter data from 18 institutions from January 2001 through June 2014, reports varying success [33]. Of a total of 370 cases of various cardiac disease, fetal intervention did not impact survival to discharge [32]. One of the key findings in the registry was the role of fetal cardiac intervention in achieving biventricular repair after birth. In fetuses with a diagnosis of aortic stenosis or evolving HLHS who underwent fetal cardiac intervention, twice as many were discharged with biventricular circulation [33].

HLHS with restrictive or intact atrial septum is another area of interest for fetal cardiac intervention. The goal of fetal intervention is to decrease left atrial pressure creating a non-restrictive atrial level shunt. The procedure of choice varies per institute and commonly involves atrial septoplasty or atrial stent placement [34]. In the International Fetal Cardiac Intervention Registry by Moon-Grady et al., 37 fetuses with HLHS underwent atrial septum intervention. A total of 24 out of 37 fetuses had successful intervention with 22/37 live births. However, mortality in the newborn period remained high and only 37 percent (14/17) survived to discharge [33].

In another report by Jantzen et al. from same registry, the rate of procedural success was 85% (N:23/27) for septoplasty and 65% for atrial stent placement (N:13/20) [34].

The procedure-related fetal demise was 13% and survival was poor, with aggregate intervened and control fetuses with survival to discharge of 35% [34]. For cases with documented follow-up, fetuses with an unrestricted foramen ovale at birth had a markedly improved survival relative to cases without intervention [34].

On the contrary, the intervention for pulmonary atresia with intact ventricular septum and critical pulmonary stenosis demonstrates less success. Among 84 cases, fetal complications were reported in 55% of cases including death and delayed fetal loss [35].

However, among successful cases, the measurement of the tricuspid valve increased on successive imaging with a higher percentage of biventricular function compared to non-intervened fetuses [35].

Complications of fetal interventions are relatively common and are often severe, including fetal hemodynamic instability manifesting as ventricular dysfunction and bradycardia [36]. An evaluation of 83 fetuses undergoing fetal intervention demonstrated hemodynamic instability in 45% of cases. This was notably associated with ventricular distortion and one case of hemopericardium [36]. Though poorly understood, it is hypothesized that this may be due to a ventricular reflex or secondary to impaired cardiac output due to ventricular distortion [36]. Improved understanding of fetal hemodynamics and electrophysiology may minimize this risk in the future. Several targets to improve the success of fetal cardiac intervention have been identified. Among these are laser technology for septal ablation, endoscopy to directly visualize the area of intervention, intraperitoneal imaging to improve visualization and miniaturization of catheters in order to enable intervention earlier in gestation than is currently possible [37].

Fetal cardiac intervention is technically challenging, and the learning curve is rather steep [38]. Using sheep as a model organism, a cardiac intervention team demonstrated significant reduction in the time required to position a needle tip at the aortic root, suggesting that simulation and repetition can improve technical aptitude [38]. For this reason, it has been suggested that single center commitment to a large-volume fetal intervention program may be the best approach to standardize the approach to these interventions and permit sufficient volume to develop technical proficiency [39].

Additional research in sheep models has evaluated the prospect of cardiac bypass for fetal intervention. Currently, this area is limited by significant morbidity due to placental dysfunction [40]. However, novel devices, including the TinyPump, have demonstrated limited success [40]. Though fetuses deteriorated after bypass on the TinyPump, the rate of compromise was slower than on a roller head pump [40]. Though clearly, further research is required, the prospect of fetal cardiac intervention on a bypass may be a reality in the decades to come. Currently, the surgical community has not been impacted by fetal intervention in dramatic ways [41]. However, as the field expands it is anticipated that physiologic changes in utero may change the approach to surgical intervention in such fetuses [41]. Further research must also target the long-term outcomes of fetal intervention, including neurodevelopment. One such study reports developmental delay in patients undergoing fetal aortic valvuloplasty, at a rate similar to patients with HLHS without intervention [42]. Interestingly, biventricular physiology was not associated with better neurodevelopmental outcomes [42].

The success of fetal intervention is dependent on technical factors including vascular access, fetal positioning and cardiac anatomy [43]. These factors likely contribute to the relatively high risk of such interventions and associated fetal and maternal morbidity and mortality [43]. Further, investigation into the genetic and molecular basis of congenital heart disease and emerging technology including stem cell research, nanotechnology and personalized medicine may allow for further advances in fetal cardiac intervention [44]. Proper case identification is imperative to assure the safety of the mother and the fetus, alike, as well as to allow for optimal results [45]. This includes a thorough echocardiographic interrogation and assessment of fetal life [45].

Importantly, fetal interventions have also demonstrated safety in mothers, with single center studies reporting no major maternal complications related to fetal cardiac interventions [46]. Among a series of 113 mothers, only one major complication, a placental abruption, was reported [46]. No mothers required intensive care and only two required tocolysis. Importantly, 89% of deliveries were at term [47]. As individual centers further develop their experience and expertise, multicenter and international collaboration is imperative to share best practices and advance the field for the benefit of all.

## 6. Conclusions

Whether pharmacologic or percutaneous, our understanding as well as the outcome for approaching congenital heart disease in the fetus has advanced by leaps and bounds. Developments in fetal imaging have improved the diagnosis and characterization of congenital heart disease in utero. However, there continues to be a need for larger studies and more experience in this highly challenging field. Collaboration between various centers to have a more uniform approach for pharmacologic intervention and a few nationalized centers with high volume for percutaneous intervention will help to achieve optimal outcomes. As these approaches increase in utilization, such collaboration may allow for the sharing of best practices and generation of guidelines to support widespread utilization. Future research aiming to improve the understanding of fetal hemodynamics and advanced embryology may direct the development of novel interventions. In addition, the expansion of pharmaceutical approaches and application in larger populations may broaden horizons for non-interventional therapies.

## Figures and Tables

**Figure 1 children-09-00812-f001:**
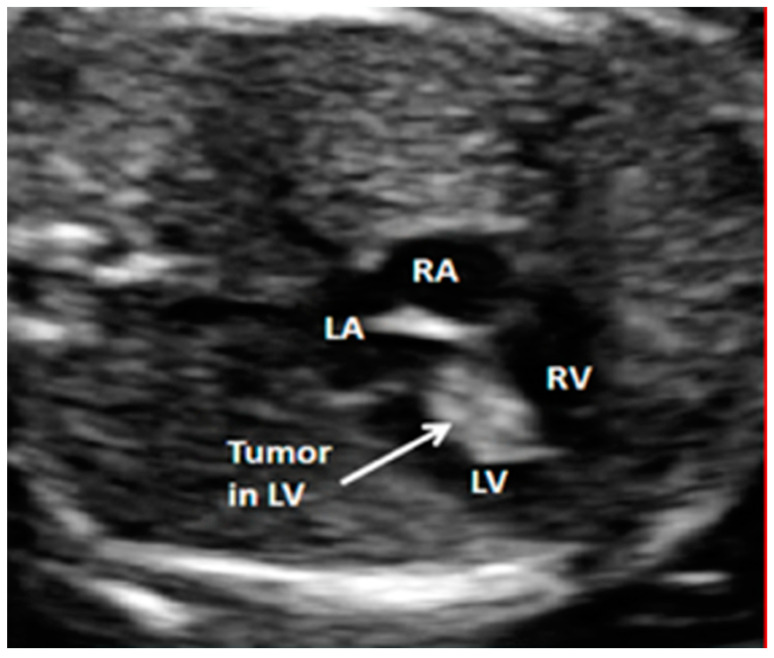
An intracardiac tumor attached to the ventricular septum as observed on four chamber view—LA, left atrium; RA, right atrium; LV, left ventricle; RV, right ventricle.

**Table 1 children-09-00812-t001:** An overview of fetal cardiac procedures.

Overview of Fetal Cardiac Procedures		
Fetal cardiac interventions	Congenital heart disease	Aortic stenosis: Balloon valvuloplasty Pulmonary stenosis: Balloon valvuloplasty Restrictive atrial septum: HLHS, D-TGA: Atrial septoplasty or atrial stent placement
Other interventions	Fetal arrhythmia	Transplacental treatment: Antiarrhythmic medications (digoxin, sotalol, flecainide or a combination); immunomodulator (intravenous immunoglobulin, steroids); sympathomimetic agents
	Congestive heart failure	Amnioreduction Pericardiocentesis Medical management—Digoxin
	Twin–twin transfusion syndrome	Amnioreduction Endoscopic laser ablation

Legend: HLHS: hypoplastic left heart syndrome; D-TGA: dextro-transposition of the great arteries; TTS: twin–twin transfusion syndrome.

**Table 2 children-09-00812-t002:** An overview of fetal cardiac diagnostic work-up.

	Summary of Fetal Cardiac Diagnostic Work-Up
Congenital heart disease	Fetal echocardiogram Fetal MRI
Fetal arrhythmia	Fetal magnetocardiogram (FMCG)
Other	Fetal hyperoxia testing

## Data Availability

The data presented in this study are openly available in reference number [1,2,3,4,5,6,7,8,9,10,11,12,13,14,15,16,17,18,19,20,21,22,23,24,25,26,27,28,29,30,31,32,33,34,35,36,37,38,39,40,41,42,43,44,45,46,47].

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
