# Peer review of "Advances in the Prenatal Management of Fetal Cardiac Disease"

_children, 2022, doi:10.3390/children9060812_

Round 1

Reviewer 1 Report

Dear Authors,

Congratulations for your work, , it is summarizing in an interesting way  various aspects of the treatment of fetal cardiovascular diseases. I do not see the need for corrections, in my opinion it can be published in this version.

Author Response

Dear Authors,

Congratulations for your work, , it is summarizing in an interesting way  various aspects of the treatment of fetal cardiovascular diseases. I do not see the need for corrections, in my opinion it can be published in this version.

Thank you for your review. Your support is appreciated.

Reviewer 2 Report

The paper  “Review of Advances in Fetal Cardiac Interventions”  wants apparently to give an overview regarding the prenatal management.

Effectively, the title of the paper  is inappropriate – it should be “ …advances in the prenatal management of fetal cardiac anomalies”,  

because, the term of Fetal cardiac interventions is currently applied specifically to the  fetal procedures with catheterization of the aortic and pulmonary valves and stenting of the interatrial septum.

Therefore, the paper should be rewritten – changing all the paragraphs where they apply in an ainappriate way the term “intervention”

  1. 27 - prenatal intervention to be changed to “approach” ….. to treat cardiac disease dates back to 1975, when Eibschitz et al reported use of propranolol to treat fetal ventricular tachycardia…

as well p.30 – “Fetal cardiac intervention (FCI) ---it is erroneous to apply this terminology to all approaches toward fetal problems, arrhythmias etc.

as in. The paragraph 3 – p. 91 – pharmacological intervention  (NO- approach)

Only the paragraph 5 – p.138 nd so on  regards the true fetal intetventions.

The previous paragraphs regard the types of fetal approach – management, in different occasions.

In conclusion: the paper should be rewritten, considering these points, starting with the abstract and adding some other key words.

Author Response

The paper  “Review of Advances in Fetal Cardiac Interventions”  wants apparently to give an overview regarding the prenatal management.

Effectively, the title of the paper  is inappropriate – it should be “ …advances in the prenatal management of fetal cardiac anomalies”,  

because, the term of Fetal cardiac interventions is currently applied specifically to the  fetal procedures with catheterization of the aortic and pulmonary valves and stenting of the interatrial septum.

Response: Thank you for this comment. The title has been amended as suggested

Therefore, the paper should be rewritten – changing all the paragraphs where they apply in an ainappriate way the term “intervention”

  1. 27 - prenatal intervention to be changed to “approach” ….. to treat cardiac disease dates back to 1975, when Eibschitz et al reported use of propranolol to treat fetal ventricular tachycardia…

Response: This statement was modified per this suggestion.

as well p.30 – “Fetal cardiac intervention (FCI) ---it is erroneous to apply this terminology to all approaches toward fetal problems, arrhythmias etc.

This statement was modified per this suggestion.

as in. The paragraph 3 – p. 91 – pharmacological intervention  (NO- approach)

Response: This statement was modified per this suggestion.

Only the paragraph 5 – p.138 nd so on  regards the true fetal intetventions.

The previous paragraphs regard the types of fetal approach – management, in different occasions.

Response: Thank you for this comment. As above, this language was corrected as noted.

In conclusion: the paper should be rewritten, considering these points, starting with the abstract and adding some other key words.

Reviewer 3 Report

hanks to the researchers for the review.
I suggest some recommendations

-According to the development of the manuscript,  you made a general description of medical therapies in fetal cardiology and some fetal intervention therapies.  
-The development of the article makes a brief description of some of these,   and does not focus on interventional therapies, for example fetal aortic valvuloplasty. So the title is not clear and objective

-Therefore, the title should be improved. For example:
"An Overview of Advances in Fetal Cardiac Medical Therapies and Interventions"
"A general Perspective on Fetal Cardiovascular Therapeutic Advances"

-Provide a brief description of the methodology of your review
-Provide article search dates (From 19.. --to 20...--)
-[MESH] terms used
-Selection of articles (Reviews, case reports, clinical and experimental studies, among others)

-Table 1 is too simple.
-You can improve it with a 3rd column where you cite some of the most relevant studies (for example: fetal arrhythmias (cite at least the author who started the treatment in column 3, describe the highlights of the study. And do it the same with each topic)

-The conclusion must be be improved. Could you give your point of view on fetal cardiovascular advances and future prospects? Highlighting the development of fetal cardiology and collaboration between specialized centers, like as you do in the last sentence

Author Response

Reviewer 3:

Thanks to the researchers for the review.
I suggest some recommendations

-According to the development of the manuscript,  you made a general description of medical therapies in fetal cardiology and some fetal intervention therapies.  
-The development of the article makes a brief description of some of these,   and does not focus on interventional therapies, for example fetal aortic valvuloplasty. So the title is not clear and objective

-Therefore, the title should be improved. For example:
"An Overview of Advances in Fetal Cardiac Medical Therapies and Interventions"
"A general Perspective on Fetal Cardiovascular Therapeutic Advances"

RESPONSE: The title has been amended to ” Advances in the Prenatal Management of Fetal Cardiac Disease” per ethis suggestion.

-Provide a brief description of the methodology of your review
-Provide article search dates (From 19.. --to 20...--)
-[MESH] terms used
-Selection of articles (Reviews, case reports, clinical and experimental studies, among others)

-Table 1 is too simple.
-You can improve it with a 3rd column where you cite some of the most relevant studies (for example: fetal arrhythmias (cite at least the author who started the treatment in column 3, describe the highlights of the study. And do it the same with each topic)

RESPONSE: A column was added highlighting recent studies in these areas. As the objective of this review is to discuss recent advances, we chose to include the most recent publications included in the review.

-The conclusion must be be improved. Could you give your point of view on fetal cardiovascular advances and future prospects? Highlighting the development of fetal cardiology and collaboration between specialized centers, like as you do in the last sentence

RESPONSE: The conclusion has been expanded to offer more recommendations for further research.

Round 2

Reviewer 2 Report

I have seen several modifications in the text, correct, however, the Table 1 and 2 need to be separated and corrected.

The Table 1  An overview of fetal cardiac interventions HLHS, hypoplastic left heart syndrome; D-TGA, Dextro-Transposition of the Great Arteries; TTS, Twin-Twin Transfusion.

It should be: An  overview of fetal cardiac procedures:

--Fetal cardiac interventions:

Congenital heart disease:  · Aortic stenosis: Balloon valvuloplasty

  • Pulmonary stenosis: Balloon valvuloplasty
  • Restrictive atrial septum: HLHS, D-TGA: Atrial septoplasty or atrial stent

                                                   placement

--- Procedures for other situations

  • Congestive heart failure…….
  • Twin-Twin Trasfusion syndrome……
  • Fetal arrhythmias……..
  •  
  • Recent Studies: can be added as it is done      McElhinney, Moon-Grady, Jantzen, Hogan, Sebastian, Schidlow

There should be a leged bellow: HLHS, hypoplastic left heart syndrome; D-TGA, Dextro-Transposition of the Great Arteries; etc.

The Table 2: Summary of Fetal cardiac Diagnostic Work.up     ---Can remain is it is, separated.…

Author Response

These tables have been modified as suggested.